# In Vivo Confocal Microscopy Findings in Corneal Stromal Dystrophies

**DOI:** 10.3390/diagnostics15020182

**Published:** 2025-01-14

**Authors:** Süleyman Okudan, Emine Tınkır Kayıtmazbatır, Ayşe Bozkurt Oflaz, Banu Bozkurt

**Affiliations:** Department of Ophthalmology, Faculty of Medicine, Selcuk University, Konya 42130, Türkiye; drtinkir@gmail.com (E.T.K.); draysebozkurtoflaz@yahoo.com (A.B.O.); drbanubozkurt@yahoo.com (B.B.)

**Keywords:** corneal stromal dystrophies, granular corneal dystrophy, macular corneal dystrophy, lattice corneal dystrophy, in vivo confocal microscopy (IVCM)

## Abstract

**Background/Objectives:** In this study, we aim to evaluate in vivo confocal microscopy (IVCM) findings of corneal stromal dystrophies (CSDs) including granular, macular and lattice corneal dystrophy that can be used for differential diagnosis and monitoring recurrences after surgical interventions. **Methods**: Patients diagnosed with CSD who were followed-up in the cornea and ocular surface unit were included in this study. IVCM was performed using the Heidelberg Retina Tomograph 3, Rostock Cornea Module (Heidelberg Engineering, Germany) and anterior segment optical coherence tomography (AS-OCT) imaging was performed using the Spectralis OCT (Heidelberg Engineering, Germany). The morphological structure, size and location of deposits, epithelial involvement and presence of inflammatory and dentritic cells were compared among the three stromal dystrophies. **Results**: A total of 72 eyes from 36 participants were included in this study. Twelve patients (33.33%) had granular corneal dystrophy (GCD), ten (27.77%) had macular corneal dystrophy (MCD) and fourteen (38.88%) had lattice corneal dystrophy (LCD). In GCD, highly reflective deposits varying in size (20 µm–300 µm) were observed. In MCD, diffuse hyperreflective stroma with dark striae, dentritic cells around deposits and abnormal keratocytes were observed. In LCD, there were branching, lattice-like and granular deposits with epithelial cell disruption in some of the eyes. In MCD, the central corneal thickness was thinner (449.44 ± 65.45 µm) compared to GCD and LCD (565.16 ± 49.62 µm and 569.91 ± 39.32 µm *p* < 0.001). Recurrence was observed in five patients following penetrating keratoplasty. **Conclusions**: IVCM is a valuable tool for distinguishing CSD subtypes and monitoring recurrence following surgical interventions.

## 1. Introduction

Corneal dystrophies (CDs) refer to a group of genetically determined bilateral, non-inflammatory diseases of the cornea. These diseases are characterized by spontaneous development of corneal opacities and often lead to loss of corneal transparency [1,2]. Groenouw first described stromal dystrophies in 1890 [3]. Corneal stromal dystrophies (CSD) are usually characterized by abnormal deposition of material in the stromal layer of the cornea. Hyaline, glycosaminoglycan and amyloid fibrils were shown to accumulate in the stroma of eyes with granular corneal dystrophy (GCD), macular corneal dystrophy (MCD) and lattice corneal dystrophy (LCD), respectively. The identification of genetic mutations has led to a better understanding of the basic defects of these diseases and a more precise diagnosis through molecular testing [2,4,5].

While corneal dystrophies may remain asymptomatic in early stages, symptomatic cases frequently present with progressive bilateral visual impairment, most commonly in the form of irregular astigmatism. Additional symptoms such as photophobia, dry eyes, corneal edema and recurrent corneal erosions may occur depending on the affected layer. Disease severity varies widely, with some cases requiring corneal transplantation due to significant vision loss and structural changes in the cornea. Among the stromal dystrophies, granular, macular and lattice corneal dystrophies are prominent and clinically distinct entities. Each exhibits unique features while sharing a common underlying mechanism of progressive corneal deposition and disruption of transparency.

GCD is marked by the development of discrete, crumb-like or flake-like deposits in the corneal stroma. Symptoms, including mild visual impairment and recurrent corneal erosions, typically emerge in the first or second decade of life. On slit-lamp examination, these opacities are interspersed with clear areas of cornea, distinguishing GCD from other dystrophies. Over time, the deposits enlarge and increase in density, often resulting in significant visual decline by the fifth decade [1].

MCD is a severe, progressive dystrophy that generally presents between 10 and 30 years of age. Patients most commonly report vision loss, with visual acuity ranging from 20/40 to 20/100 or worse, depending on the density and distribution of deposits. Symptoms such as photophobia, irritation and pain can occur but are less frequent and are often linked to recurrent corneal erosions. Examination reveals diffuse stromal haze and irregular whitish deposits extending from limbus to limbus, without clear areas between deposits. Advanced stages are characterized by corneal thinning, involvement of Descemet’s membrane and endothelial dysfunction, necessitating surgical intervention in many cases [6].

LCD typically manifests in the first or second decade of life with recurrent corneal erosions and subepithelial deposits. As the disease progresses, the deposits develop into lattice-like, refractile branching lines with dichotomous endings. These changes are accompanied by stromal haze and corneal irregularities, leading to significant visual impairment. Unlike GCD and MCD, the peripheral cornea and Descemet’s membrane are generally spared in early disease [7].

Timely identification of CD through clinical examination and genetic testing is crucial for optimizing outcomes. Treatment options range from conservative measures, such as lubricants and therapeutic lenses, to surgical interventions, including phototherapeutic keratectomy (PTK) and corneal transplantation. Improved understanding of the genetic and clinical diversity of these dystrophies is essential for tailoring treatment strategies and achieving better outcomes for affected individuals.

In vivo confocal microscopy (IVCM) is a non-invasive technology that provides high-resolution images of corneal microstructures at cellular levels, offering unique advantages over traditional methods for diagnosing and monitoring corneal diseases [8,9,10,11]. There are many studies about the use of IVCM in CD, refractive surgery, dry eye and meibomian gland disease and infectious keratitis [12,13,14]. In CD, this technique allows detailed examination of the size, morphology and place of abnormal deposits and associated cellular and nerve fiber changes. Therefore, IVCM has become an important tool in the diagnosis and progression of CD, and treatment responses. While conventional diagnostic methods such as slit lamp biomicroscopy can often detect changes that become apparent in the advanced stages of the disease, IVCM has the ability to identify microscopic deposits and microstructural changes at earlier stages [15,16].

In this study, we aim to investigate the distinctive IVCM features of deposits and cellular changes in GCD, MCD and LCD that can be used for differential diagnosis and monitoring recurrences after surgical interventions.

## 2. Materials and Methods

Patients diagnosed with CSD who were followed up in the cornea and ocular surface unit were included in this study. Demographic data, including age, gender, medical and family history, were recorded. High-resolution corneal imaging was conducted using the IVCM Heidelberg Retina Tomograph 3, Rostock Cornea Module (Heidelberg Engineering, Germany) by the same operator (BB). This device operates on a reflectance-based imaging system, utilizing laser scanning technology to analyze light reflected from the tissue. It provides high-resolution and detailed images of corneal microstructures without the need for fluorescence. After instilling 0.5% topical proparacaine hydrochloride (Alcaine^®^, Alcon Pharmaceuticals, Bornem, Belgium), a high-viscosity contact gel ((Viscotears Gel ophthalmic ointment; Novartis Pharma AG, Basel, Switzerland) was applied to the lower conjunctival fornix and the front surface of the microscope lens (Zeiss x63). The microscope’s objective lens was equipped with a sterile, single-use polymethylmethacrylate cap (TomoCap). Participants were seated comfortably with their chin and forehead supported and were instructed to focus on the center of the objective lens. To ensure proper positioning, a lateral view of the eye and objective lens was captured with a digital camera for each scan. The x–y image alignment and section depth were manually adjusted. Multiple corneal images were obtained within five minutes, with no complications associated with the confocal microscopy procedure.

Anterior segment optical coherence tomography (AS-OCT) imaging was performed using the Spectralis AS-OCT system (Heidelberg Engineering GmbH, Heidelberg, Germany). This high-resolution device utilizes a wavelength of 870 nm to provide detailed cross-sectional images of the cornea and anterior segment structures. The imaging protocol included the acquisition of both horizontal and vertical cross-sectional scans centered on the corneal apex. Before imaging, patients were positioned comfortably with their chin and forehead supported in the device’s headrest to ensure stability and alignment. The built-in infrared fixation target was used to maintain the patient’s gaze and minimize movement during scanning. The imaging parameters, including scan density, depth and resolution, were adjusted as needed to achieve optimal visualization of the corneal layers. The corneal epithelial layer, stromal deposits and posterior segment structures were evaluated in all patients. Images were reviewed for stromal hyperreflectivity, presence and localization of deposits and structural changes and thickness of the cornea. All scans were conducted by a single experienced operator to minimize variability. The imaging process was non-invasive, painless and completed within a few minutes. No complications or adverse effects were observed during or after the procedure.

The demographic data, morphological structure, size and location of deposits, epithelial involvement and presence of inflammatory and dentritic cells observed in IVCM sections and AS-OCT findings were compared among GCD, MCD and LCD.

### Statistical Analysis

The statistical analysis was performed using the SPSS software package (version 18.0). Descriptive statistics were expressed as numbers (n) and percentages (%) for categorical variables, and as mean ± standard deviation (SD) for continuous variables. The chi-square test and Fisher’s exact chi-square test were employed to compare categorical variables. The Kolmogorov–Smirnov test was used to assess the normality of numerical data. A *p*-value of less than 0.05 was considered statistically significant, and results were evaluated within a 95% confidence interval.

## 3. Results

A total of 72 eyes from 36 participants were included in this study. Among the patients, 17 (47.22%) were female and 19 (52.77%) were male. The mean age was 40.97 ± 18,22 years (min 12–max 76 years). Of the patients, 12 (33.33%) had GCD (Figure 1a), 10 (27.77%) had MCD (Figure 1b) and 14 (38.88%) had LCD (Figure 1c).

The age distribution among the dystrophy types revealed that individuals with GCD had the highest mean age (49.92 ± 19.55), followed by those with LCD (40.14 ± 15.74), while individuals with MCD exhibited the lowest mean age (31.4 ± 14.47). Statistically significant differences in age were observed among the three dystrophy types (*p* < 0.05 for all pairwise comparisons).

Notably, 14 patients had a positive family history, emphasizing the genetic inheritance pattern of these dystrophies. Genetic mutations were identified in 16 patients. TGFBI gene mutation has been reported in all patients with GCD. 

IVCM of eyes with GCD revealed round, highly reflective particles forming bigger, grapelike deposits throughout the stromal layer (Figure 2a). The deposit diameter was changing (20 µm–300 µm). A weak negative correlation was observed between age and depth (ρ = −0.165, *p* = 0.541). The stroma between the deposits remained relatively transparent, indicating areas of preserved tissue integrity (Figure 2a). Keratocytes appeared normal, with no signs of significant remodeling or cellular disruption. Epithelial involvement was observed in eight cases (66.66%) (Figure 3a), while the endothelial layer could not be visualized in nine patients. Two patients had recurrent erosions. In AS-OCT, highly reflective, discrete stromal deposits were observed mainly in the anterior and middle stromal layer, but also in the posterior stromal layer extending to the endothelium in some subjects (Figure 4a). 

In MCD, IVCM revealed diffuse hyperreflective deposit accumulation throughout the whole stroma with characteristics dark striae inside (Figure 2b). The stromal sections showed a notable loss of keratocyte visibility, reflecting substantial stromal remodeling and disorganization. Epithelial involvement was observed in four cases (Figure 3b), while the endothelial layer could be visualized only in one patient. In some cases, round inflammatory cells and dendritiform cell clusters around the deposits and amorphous keratocytes were observed (Figure 5a,b). AS-OCT revealed a diffuse increase in stromal reflectivity extending from the Bowman to the endothelial layer and limbus to limbus (Figure 4b). 

LCD exhibited distinct structural changes and epithelial disruption on IVCM. The hyperreflective deposits formed a branching, lattice-like pattern and were predominantly located in the stroma extending to the epithelium and, unlike MCD, the keratocytes were clearly visible between the deposits (Figure 2c). Epithelial disruption was a prominent feature, contributing to the recurrent epithelial erosions frequently associated with this dystrophy (Figure 3c). In patients with LCD, epithelial involvement was observed in 10 cases, while the endothelial layer could be visualized in 4 cases. Eleven eyes of six patients with LCD had recurrent corneal erosions. AS-OCT imaging supported these observations by demonstrating round and linear deposits mostly in the anterior and middle stroma. (Figure 4c). 

The mean central corneal thickness (CCT) values for the three groups, excluding patients with prior corneal surgery, were as follows: 565.16 ± 49.62 µm for GCD (Figure 4a), 449.44 ± 65.45 µm for MCD (Figure 4b) and 569.91 ± 39.32 2 µm for LCD (Figure 4c). A one-way ANOVA test revealed a statistically significant difference in CCT among the three groups (*p* < 0.001). These findings indicate that MCD had a notably thinner corneal thickness compared to the other dystrophies. 

In our study, one eye of a patient with GCD and one eye of a patient with MCD underwent penetrating keratoplasty (PKP). Recurrence was observed in the GCD patient. Among six patients who underwent bilateral PKP (2 GCD, 2 MCD and 2 LCD), one GCD patient, one LCD patient and both MCD patients experienced recurrence (Figure 6a,b), while the other GCD patient developed graft rejection. Additionally, bilateral phototherapeutic keratectomy (PTK) was performed in two patients with GCD and two patients with LCD. Recurrence occurred in one GCD patient and one LCD patient following PTK (Figure 7a,b).

## 4. Discussion

In this study, we demonstrated IVCM findings of granular-, macular- and lattice-type CSD. In our previous paper, R555W mutation in exon 12 of the TGFBI gene mutation was shown in all patients with GCD [4]. Our findings align with prior literature, confirming the broad distribution of hyaline deposits throughout the stromal layers and in the epithelium in some sections [17]. Notably, these deposits varied in size 20 µm–300 µm and were distributed from the anterior stroma to the endothelium. Werner et al. [18] provided a comprehensive analysis of GCD deposits using IVCM, highlighting their structural variability across different corneal layers. In their study, irregular, highly reflective “breadcrumb-like” deposits, approximately 50 µm in diameter, were observed between the intermediate epithelial cell layer and Bowman’s layer. Stromal deposits exhibited a range of shapes, including round, irregular and trapezoidal, with sizes varying from 50 to 500 µm. These deposits were highly reflective, dense, gray-white and separated by clear (dark) areas of normal stroma.

Additionally, smaller trapezoidal deposits (30 to 50 µm) were occasionally identified in the deeper stroma, surrounded by normal keratocytes. The deep stroma also contained multiple, highly reflective punctiform deposits (5 to 10 µm), interspersed with round structures likely corresponding to keratocyte nuclei. Werner et al. also noted that deposits in the superficial corneal layers appeared flat, whereas those in the deeper layers displayed a more three-dimensional form [18]. These findings emphasize the multi-layered involvement in GCD, showcasing the variable morphology and distribution of deposits. The varying deposit diameters observed in our study corresponds to findings in this study emphasizing differences in deposit size and distribution. This observation underscores the progressive nature of GCD, with deposits appearing initially in the anterior stroma and spreading and combining over time. IVCM findings showed no significant keratocyte remodeling or cellular disruption, suggesting that granular dystrophy’s impact on stromal architecture might be less severe in its early stages. These findings highlight the broad distribution of hyaline deposits and the overall preservation of stromal architecture, which aligns with the typically milder impact of GCD on vision until advanced stages. This is consistent with the typically mild clinical symptoms of GCD, with patients often remaining asymptomatic until moderate–advanced stages when coalescence and deeper stromal involvement occur. The stroma between the deposits remained relatively transparent, indicating preserved tissue integrity. However, in cases where deposits extended to the epithelium, the risk of recurrent erosions increased, correlating with the clinical symptoms of pain and discomfort reported in advanced cases as reported in prior studies [19,20]. Dalton et al. reported that on IVCM, images of the epithelium and anterior stroma in GCD demonstrate central corneal irregularity due to the accumulation of “breadcrumb-like” hyaline deposits [21]. In this study, epithelial involvement was observed in 8 out of 12 cases of GCD; two of whom had a history of recurrent corneal erosions. Bilateral PTK was performed in two patients for recurrent corneal erosions. In our study, AS-OCT findings revealed highly reflective stromal deposits forming isolated islands with localized stromal thickening. This imaging modality further supports the role of advanced diagnostic tools in characterizing GCD, as OCT enables precise localization and depth assessment of deposits [22]. The localized thickening observed with AS-OCT corresponds to the stromal deposits, emphasizing the utility of combining IVCM and OCT in the diagnostic workup of stromal dystrophies. Although GCD typically spares the endothelial layer, we found that the endothelial layer could not be visualized in 9 out of 13 cases, potentially indicating advanced disease stages or limitations in imaging through dense stromal opacities. This finding warrants further investigation to understand its clinical implications and whether it reflects a true pathological process or technical challenges in imaging. Among the patients with GCD, two underwent bilateral PKP, with one experiencing recurrence and the other developing graft rejection, while one patient who underwent unilateral PKP also experienced recurrence.

Regarding the MCD, IVCM findings revealed increased stromal hyperreflectivity, tissue disorganization and a notable loss of keratocyte visibility. These observations are consistent with the pathological mechanisms of MCD, as previous studies have demonstrated that metabolic abnormalities in glycosaminoglycan synthesis result in keratocyte dysfunction and stromal remodeling. The diffuse deposition of unsulfated keratan sulfate, along with increased chondroitin 6-sulfate, disrupts the stromal architecture, leading to progressive corneal opacification and visual impairment. This metabolic dysfunction is a hallmark of MCD and explains the severe disruption observed in stromal layers [23]. In our study, epithelial involvement was observed in four patients with MCD, consistent with the relatively preserved epithelial layer and diffuse light reflective materials with dark stria-like images througout the anterior stroma reported in the literature [16,24]. Additionally, dendritiform cell clusters and amorphous keratocytes were observed in some cases, suggesting a chronic inflammatory process and active keratocytes playing role in the biosynthesis of keratan sulphate that may contribute to the progression of corneal opacification. We were able to visualize the endothelial layer in two patients and observed small keratic presipitates, while the remaining cases showed insufficient clarity, likely due to dense stromal deposits obscuring deeper layers. This observation underscores the severity of stromal involvement in MCD and highlights the potential limitations of imaging in advanced stages of the disease. The absence of endothelial visualization in most cases does not necessarily indicate endothelial dysfunction but may reflect the technical challenges of penetrating thick stromal deposits during imaging. AS-OCT findings in our study further reinforced the diffuse involvement of the cornea in MCD. Diffuse hyperreflectivity was visualized throughout the stroma, along with significant corneal thinning and reduced central corneal thickness (449.44 ± 65.45 µm). These findings align with previous studies reporting lower CCT and smaller corneal volume in MCD compared to other stromal dystrophies [25,26]. Such imaging features are invaluable for distinguishing MCD from similar conditions and are critical for surgical planning, particularly when determining the suitability of PKP versus deep anterior lamellar keratoplasty (DALK) [27]. Among the patients with macular corneal dystrophy (MCD), one underwent unilateral PKP, while two underwent bilateral PKP. Recurrence was observed in both patients who underwent bilateral PKP.

IVCM findings of LCD revealed branching, highly reflective branching deposits primarily located in the superficial and middle stroma. These deposits exhibited irregular, lattice-like patterns with indistinct margins. Despite the presence of these deposits, the surrounding stromal tissue appeared relatively preserved, with visible keratocytes in the areas between the deposits. Epithelial involvement was a prominent feature in our study, with IVCM demonstrating epithelial irregularities in most cases. These epithelial changes are closely associated with the recurrent erosions frequently reported in LCD [28]. In our study, 11 eyes of six patients with LCD exhibited recurrent corneal erosions, emphasizing the impact of progressive anterior corneal involvement caused by underlying stromal deposits on clinical symptoms such as irritation, pain and visual impairment. AS-OCT findings supported the observations made with IVCM, showing localized irregularities in the anterior and middle stroma and disruptions in the Bowman layer. These findings align with previous studies that demonstrate the neurotropic phenomenon in LCD, where the gradual accumulation of amyloid deposits can progressively wrap corneal nerve fibers, contributing to disease progression and symptom severity [29]. Rosenberg et al. [30] conducted an in-depth analysis of corneal morphology in LCD type 2 (Familial Amyloidosis, Finnish Type) and reported several findings. Among 20 examined corneas, 1 exhibited an epithelial ridge, while basal epithelial pleomorphism was observed in 5 cases. Dense deposits were present in 13 corneas, and severe fibrosis was noted in 5 corneas. Thick anterior and middle stromal filaments, corresponding to lattice lines visible on biomicroscopy, were identified in 15 corneas, while undulated stromal structures were observed in 11 corneas. Regarding the endothelium, it appeared normal in 17 cases, while 1 cornea showed small endothelial pits and another displayed precipitate-like aggregations. The endothelial layer was visualized in only a few cases in our cohort, likely due to the dense stromal deposits obscuring deeper layers. The absence of significant endothelial changes aligns with previous reports indicating that LCD primarily involves the anterior and middle corneal layers, with minimal impact on the endothelium [31]. Clinically, the recurrent epithelial erosions and progressive vision loss associated with LCD often necessitate surgical intervention. PTK can provide temporary relief by reducing irregular astigmatism and smoothing the corneal surface, while deeper interventions such as DALK and PKP may be required in advanced cases [32,33,34,35]. In our study, two patients with LCD underwent bilateral PTK and two patients had bilateral PKP. After PKP, recurrence was observed in one eye of a patient with LCD. IVCM plays a crucial role in monitoring disease progression and assessing the effectiveness of these treatments by enabling detailed visualization of the corneal microstructure. It has been reported that in atypical cases, early stages of the disease or instances of recurrence following treatment where the characteristic linear pattern of amyloid deposition may not be clinically apparent, IVCM can facilitate rapid recognition of the condition by identifying the distinctive reflectivity of the amyloid substance [36,37].

Our results are consistent with prior research in demonstrating the recurrence potential of CSD after keratoplasty, while also highlighting distinct patterns influenced by variations in patient demographics, genetic profiles and surgical methods [38,39]. The comparatively higher recurrence rates observed in MCD in this study, compared to previous reports, may be attributable to differences in follow-up duration or patient characteristics, emphasizing the variability in recurrence patterns even within the same subtype. These observations reinforce the need for personalized management strategies and extended follow-up to achieve better outcomes in these genetically driven conditions.

GCD, MCD and LCD are hereditary conditions that pose significant diagnostic and management challenges due to their progressive nature and impact on corneal transparency. This study highlights the critical role of IVCM in advancing the understanding and evaluation of these dystrophies. As a high-resolution, non-invasive imaging modality, IVCM provides detailed visualization of the microstructural changes within the cornea, enabling precise differentiation of dystrophy subtypes and monitoring disease progression.

Using IVCM, distinctive features of GCD, MCD and LCD were identified, reflecting the unique pathological characteristics of each dystrophy. In GCD, highly reflective, discrete deposits resembling grape-like clusters were observed, predominantly located in the anterior and middle stromal layers. These deposits are interspersed with areas of relatively preserved stromal transparency, aligning with the typically milder clinical presentation of GCD in its early stages. By contrast, MCD demonstrated diffuse stromal hyperreflectivity, significant keratocyte loss and dark striae, consistent with severe stromal remodeling and metabolic dysfunction. These findings correlate with the aggressive clinical progression of MCD, which often leads to profound visual impairment. LCD, on the other hand, exhibited characteristic lattice-like branching deposits within the stromal layers, frequently accompanied by epithelial disruptions. These changes are closely associated with recurrent corneal erosions and visual decline in affected individuals.

IVCM’s ability to detect microscopic deposits and assess cellular abnormalities provides invaluable insights into the pathophysiology of these dystrophies. The identification of inflammatory cells, keratocyte changes and stromal remodeling patterns enhances our understanding of disease mechanisms and progression. Furthermore, the visualization of epithelial involvement in conditions such as LCD and GCD highlights the clinical relevance of IVCM in diagnosing and managing recurrent corneal erosions, which is a common complication in these dystrophies.

The utility of IVCM extends beyond initial diagnosis to include the monitoring of disease progression and assessment of post-surgical outcomes. In cases of PKP or PTK, IVCM allows for the early detection of recurrent deposits, enabling timely reintervention and improved management strategies. This capability is particularly critical in MCD, where recurrence rates tend to be higher and can significantly impact long-term visual outcomes. Moreover, the ability of IVCM to provide detailed, layer-specific imaging offers significant advantages in clinical practice. Unlike traditional diagnostic methods, which often detect changes only in advanced stages, IVCM can identify early, subtle alterations in corneal structure. This early detection capability supports proactive management strategies, potentially delaying the need for invasive surgical interventions.

Despite its strengths, this study acknowledges certain limitations in the application of IVCM. The reliance on operator expertise and the need for standardized imaging protocols can introduce variability in image acquisition and interpretation. Additionally, while IVCM provides unparalleled resolution of corneal microstructures, its field of view is limited, necessitating complementary techniques for comprehensive corneal evaluation. Future advancements in imaging technology, such as automated analysis and integration with artificial intelligence, may address these limitations, enhancing the diagnostic accuracy and clinical utility of IVCM.

In conclusion, this study underscores the pivotal role of IVCM in the evaluation and management of CSD. By offering high-resolution, non-invasive insights into corneal microstructures, IVCM facilitates accurate diagnosis, detailed monitoring and early detection of disease recurrence. The adoption of IVCM in routine clinical practice represents a significant advancement in the personalized management of CD, ultimately improving visual outcomes and quality of life for affected patients. Continued research into the application of IVCM, particularly in combination with genetic and molecular studies, will further enhance our understanding of these complex conditions and pave the way for innovative therapeutic strategies.

## Figures and Tables

**Figure 1 diagnostics-15-00182-f001:**
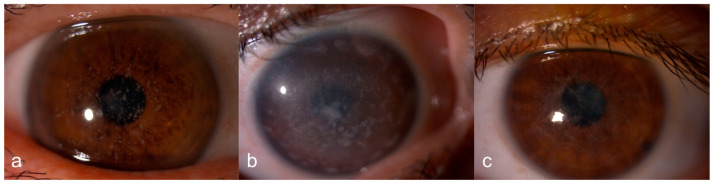
Anterior segment photographs of patients with stromal corneal dystrophies. (**a**) Granular corneal dystrophy (GCD): Discrete, well-defined opacities. (**b**) Macular corneal dystrophy (MCD): Diffuse stromal haze and ill-defined opacities. (**c**) Lattice corneal dystrophy (LCD): Branching lattice lines.

**Figure 2 diagnostics-15-00182-f002:**
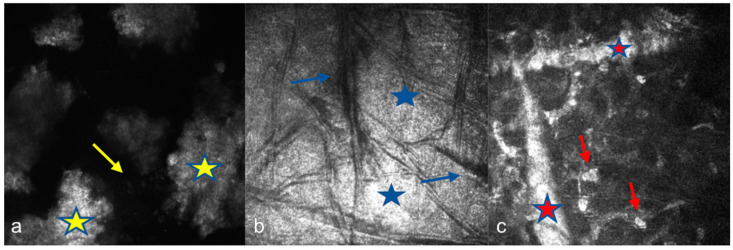
In vivo confocal microscopy (IVCM) images of corneal stromal dystrophies. (**a**) Granular corneal dystrophy (GCD): Round, highly reflective particles forming grapelike deposits (yellow stars) throughout the stromal layer with variable diameters The stroma between deposits remains relatively transparent (yellow arrow). (**b**) Macular corneal dystrophy (MCD): Diffuse hyperreflective deposit accumulation throughout the stroma (blue stars), with dark striae (blue arrows) inside and significant stromal remodeling, resulting in keratocyte invisibility. (**c**) Lattice corneal dystrophy (LCD): Branching, lattice-like hyperreflective deposits in the stroma (red stars), with visible keratocytes (red arrows) between the deposits.

**Figure 3 diagnostics-15-00182-f003:**
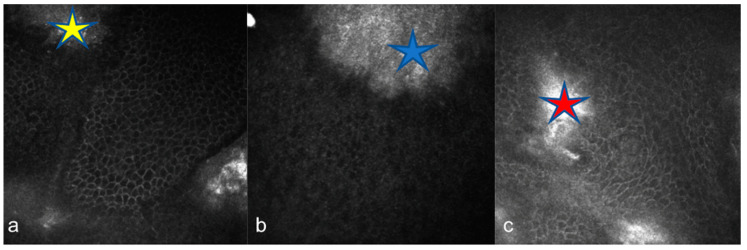
Epithelial involvement observed via in vivo confocal microscopy (IVCM) in corneal stromal dystrophies. (**a**) Granular corneal dystrophy (GCD): Epithelial involvement was observed in eight cases (yellow star). (**b**) Macular corneal dystrophy (MCD): Epithelial involvement was observed in four cases (blue star). (**c**) Lattice corneal dystrophy (LCD): Prominent epithelial disruption contributing to recurrent corneal erosions frequently associated with this dystrophy (red star).

**Figure 4 diagnostics-15-00182-f004:**
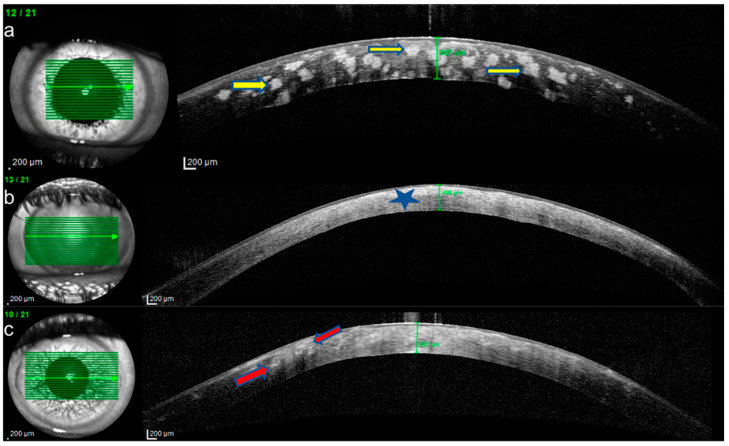
Anterior segment optical coherence tomography (AS-OCT) imaging of corneal stromal dystrophies. (**a**) Granular corneal dystrophy (GCD): Highly reflective, discrete stromal deposits primarily located in the anterior and middle stromal layers, with extension to the posterior stroma and endothelium (yellow arrows). (**b**) Macular corneal dystrophy (MCD): Diffuse limbus to limbus increase in stromal reflectivity (blue star) extending from the Bowman to the endothelial layer and thinner corneal thickness (**c**) Lattice corneal dystrophy (LCD): Round and linear hyperreflective deposits (red arrows) observed predominantly in the anterior and middle stroma.

**Figure 5 diagnostics-15-00182-f005:**
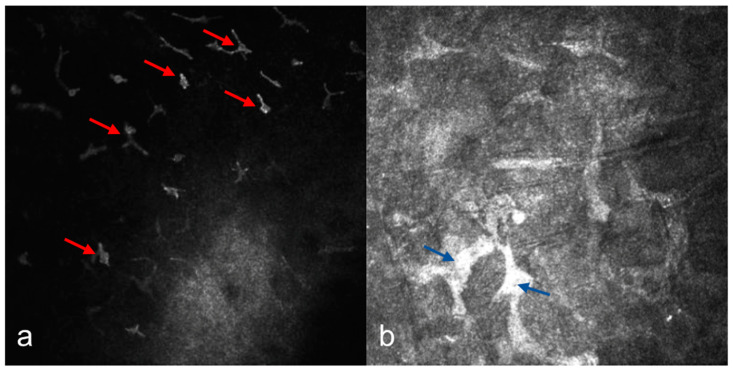
(**a**) Dendritiform cell clusters (red arrows). (**b**) Amorphous keratocytes (blue arrows) in macular corneal dystrophy (MCD).

**Figure 6 diagnostics-15-00182-f006:**
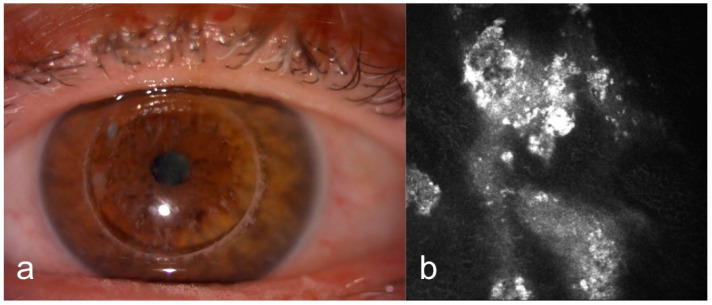
(**a**) Anterior segment photograph of a patient with GCD, demonstrating deposit accumulation on the graft following PKP. (**b**) IVCM image of the same patient, showing deposit accumulation on the graft surface.

**Figure 7 diagnostics-15-00182-f007:**
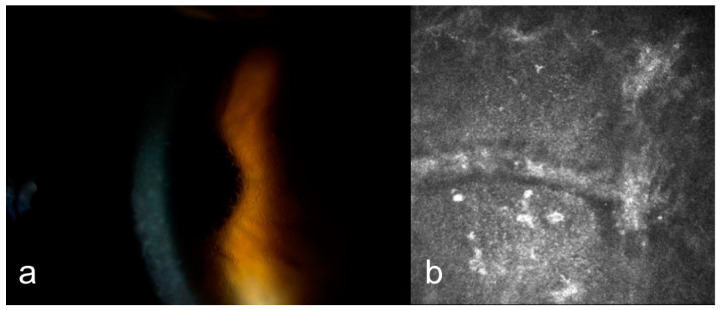
(**a**) Anterior segment photograph of a patient with LCD, demonstrating deposit accumulation on the corneal surface following PTK. (**b**) IVCM image of the same patient, showing linear lattice deposits formation after PTK.

## Data Availability

Data is contained within the article.

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
