# Peer review of "In Vivo Confocal Microscopy Findings in Corneal Stromal Dystrophies"

_diagnostics, 2025, doi:10.3390/diagnostics15020182_

Round 1

Reviewer 1 Report

Comments and Suggestions for Authors

-IVCM findings have been previously published in the literature in these dystrophies. Comparisons with the results in those studies have not been made sufficiently.

-Including IVCM images of patients who had recurrence after PKP will make the mentioned comparison visible.

-Providing IVCM images after PKT will also make the study more interesting.

Author Response

Comment 1:

IVCM findings have been previously published in the literature in these dystrophies. Comparisons with the results in those studies have not been made sufficiently.

Response 1:

Thank you for this insightful comment. We agree that the comparisons with previously published IVCM findings need to be addressed in greater detail. Accordingly, we have revised the discussion section to provide a more detailed comparison of the IVCM findings for each dystrophy with those reported in the existing literature. The revisions can be found on page [7], lines [278-281]; page [8], lines [282-290, 304-306]; page [9], lines [362-370].

Comments 2:

Including IVCM images of patients who had recurrence after PKP will make the mentioned comparison visible.

Response 2:

Thank you for this valuable suggestion. We have included IVCM and anterior segment photographs of a patient with GCD who developed recurrence after PKP to provide a visual representation of the comparison. These images have been added as Figure 6a and Figure 6b in the manuscript. The revisions can be found on page [7], lines [255-261].

Comments 3:

Providing IVCM images after PKT will also make the study more interesting.

Response 3:

Thank you for your suggestion. We have added IVCM and anterior segment photographs of a patient with LCD who developed recurrence after PKT to enrich the study and illustrate the findings more comprehensively. These images have been included as Figure 7a and Figure 7b in the manuscript. The revisions can be found on page [7], lines [263-270].

Reviewer 2 Report

Comments and Suggestions for Authors

In this manuscript, authors have demonstrated the potential of in-vivo confocal microscopy (IVCM) for the investigation of corneal stromal dystrophies (CSD) including granular, macular and lattice corneal dystrophy. They performed studies on 36 human subjects having different corneal dystrophies such as GCD, LCG and MCD. Authors found that in MCD, corneal thickness (center) was thinner relative to GCD and LCD. Authors have envisaged that IVCM as IVCM is a valuable tool for distinguishing CSD subtypes and monitoring recurrence following surgical interventions. This is a nice work and the manuscript is written well. However, authors are advised to incorporate the following changes to enhance the quality of their work.

1. Please mention clearly that the IVCM detailed in the work is based on fluroscence or reflectance based imaging ?

2. Authors need to point out (using colored arrow or so) what they are trying to show in the images Fig 2-5.  They several morphological/structural  features seen in the images. Authors need to specifically explain (pointing in Fig) what does each feature (strcutures) indicate ?

3. It is better to add a general comment on general application of laser scanning confocal microscopy in ophthalmology. In vivo and ex vivo Confocal microscopy extensively have been applied on clinical and preclinical ophthalmology to understand corneal and retinal diseases[1-3]. Authors can add this statement in the text with suggested good references.

(1) https://doi.org/10.1167/iovs.61.13.1

(2) https://doi.org/10.1136/bjo.87.2.225

(3) https://doi.org/10.4103/IJO.IJO_3013_22

4.  Do the figures 3 and 4 taken from same eye and region  of cornea? Authors have to provide this information. Need to show the IVCM and OCT images of the same patch of cornea showing the dystrophies (for thress dystrophiescases)

Author Response

Thank you very much for taking the time to review this manuscript.

Comment 1: Please mention clearly that the IVCM detailed in the work is based on fluroscence or reflectance based imaging ?

Response 1: Thank you for pointing this out. We have clarified that the IVCM used in this study operates on a reflectance-based imaging system. The following sentence has been added to the Methodology section:

Updated text in the manuscript: "This device operates on a reflectance-based imaging system, utilizing laser scanning technology to analyze light reflected from the tissue. It provides high-resolution and detailed images of corneal microstructures without the need for fluorescence." This revision can be found on page [3], lines [94-96].

Comment 2:  Authors need to point out (using colored arrow or so) what they are trying to show in the images Fig 2-5.  They several morphological/structural  features seen in the images. Authors need to specifically explain (pointing in Fig) what does each feature (strcutures) indicate ?

Response 2: 

Thank you for this valuable suggestion. We have revised Figures 2–5 to include colored arrows and stars to clearly point out and label the morphological/structural features observed in the images. Additionally, detailed explanations of what each feature represents have been added to the figure legends for clarity. These revisions can be found in the updated figures and corresponding legends Figure 2 on page [5], lines [195-205]; Figure 3, page [5], lines [209-216]; Figure 4, page [6], lines [228-238]; Figure 5, page [6], lines [239-245].

Comment 3:

It is better to add a general comment on general application of laser scanning confocal microscopy in ophthalmology. In vivo and ex vivo Confocal microscopy extensively have been applied on clinical and preclinical ophthalmology to understand corneal and retinal diseases[1-3]. Authors can add this statement in the text with suggested good references.

Response 3: 

Thank you for this suggestion. We have incorporated a general statement on the application of laser scanning confocal microscopy in ophthalmology in the Introduction section. Specifically, we utilized references [2] and [3] to support the statement, as these align closely with the scope of the study. However, we did not use reference [1] as it focuses on fluorescence-based imaging, which is outside the scope of our reflectance-based imaging methodology.

These revisions can be found in page [2], lines [76-85]

Comment 4:

Do the figures 3 and 4 taken from same eye and region  of cornea? Authors have to provide this information. Need to show the IVCM and OCT images of the same patch of cornea showing the dystrophies (for thress dystrophiescases).

Response 4:

Thank you for this observation. Figure 3 illustrates the epithelial layer of the cornea, while Figure 4 presents the corresponding OCT images from the same patients. To address the comment further, we have added IVCM and OCT images of the same patch of the cornea for three dystrophic cases, ensuring that the dystrophies are demonstrated clearly. The images in Figures 2, 3, and 4 represent the same cornea, highlighting the corresponding dystrophic features through IVCM and OCT.

Round 2

Reviewer 1 Report

Comments and Suggestions for Authors

Thank you for adressing the comments, it is much better with this version.